Proceedings of the 6th Symposium on Advances in Approximate Bayesian Inference, 2024 1–9

# Estimating Expectations without Sampling:
# Neural Stein Estimation

**Mohsin Hasan**[1]                                    MOHSIN.HASAN@MILA.QUEBEC
**Dinghuai Zhang**[1]                                  DINGHUAI.ZHANG@MILA.QUEBEC
**Cheikh Ahmed**[2,3]                        CHEIKH-ABDALLAHI.AHMED@POLYMTL.CA
**Awa Khouna**[2,3]                                   AWA.KHOUNA@POLYMTL.CA
**Yoshua Bengio**[1]                                YOSHUA.BENGIO@MILA.QUEBEC
[1] *Mila - Québec AI Institute*    [2] *Polytechnique Montréal*
[3] *École Polytechnique*

## Abstract

We propose a method for estimating the expected value of a given function $h(x)$, under an intractable distribution $p(x)$ whose score function $\nabla \log p(x)$ is however available, without sampling from it. Monte Carlo based sampling methods, in particular Markov Chain Monte Carlo (MCMC) methods when an exact sampler is not available, are a popular tool for this task. However, they may be difficult to diagnose and suffer from noisy estimates, while potentially being very expensive and biased, in the MCMC case. Our proposed method, Neural Stein Estimation (NSE), avoids these issues and instead frames calculating the expectation as solving a differential equation inspired by Stein's method and control variates. The algorithm consists of solving this differential equation through optimization, using a neural network. This means the method is deterministic and converges in a stable way, transforming the issue of sampling at run-time to the issue of sampling at training-time and the amortized cost of training this neural network. This work presents the theoretical foundations of NSE, and evaluates the method's viability over control variate baselines on simple distributions.

## 1. Introduction

In the fields of statistical inference and probabilistic modeling, the accurate estimation of expectations plays a crucial role, especially in areas like machine learning, physics, and computational biology. The expectation of the function $h$ under the distribution $p$ is:

$$E_p[h(X)] = \int h(x)p(x)\,dx$$

Computing this integral can be challenging, especially in scenarios involving high-dimensional spaces or complex distributions. Traditional methods, such as Monte Carlo (MC) techniques (Robert and Casella, 2010), tackle this by using sampling. These methods are useful but often require significant computational resources and can be inefficient, particularly with complex, high-dimensional distributions. A particular problem is that these are stochastic, which makes them difficult to diagnose and evaluate. This motivates us to explore new approaches for estimating expectations with the key goal that they have less variance and are more accurate.

A natural starting point towards this goal are control variate (CV) methods, which typically involve training an additional function $s_\theta(x)$ (the "control variate"), with parameters $\theta$, such that the alternative integrand $h(x) + s_\theta(x)$ is an unbiased estimator for $\mathbb{E}[h(X)]$ with less variance (Assaraf and Caffarel, 1999; Oates et al., 2017). This is typically done as a post-processing step once we have already obtained samples from $p$, in order to improve MC estimates.

An expressive class for $s_\theta$ are the "zero variance CVs", inspired by Stein's method, and requiring the computation of the score function $\nabla \log p(x)$ (Assaraf and Caffarel, 1999; Gorham and Mackey, 2015). If these CVs are parameterized flexibly enough (such as with a neural network) to reduce the variance completely, we can consider other loss functions with the same minima. This results in a different approach for expectation estimation which we call Neural Stein Estimation (NSE).

In contrast to MC methods, the key advantage of the method is that it doesn't require sampling, which means the approach produces stable estimates, and is easier to diagnose (using a loss function). These properties can be helpful when dealing with complex, high-dimensional distributions, where direct sampling can be challenging. The trade-off is the added cost of training the network, and that the method will estimate the expectation of a particular function $h(x)$ (and not a means of estimating multiple expectations, as in sampling).

We review the relevant background concerning CVs in section 2, followed by a description of our method in section 3. In section 4, we present some simple experimental results comparing the method to other CV methods, and in particular demonstrate that when the CV training data is not sampled from the target distribution, our method outperforms them. We conclude with a discussion of our findings and future research directions.

## 2. Background and Related Works

We denote the target distribution $p(x) = \tilde{p}(x)/Z$, with $x \in \mathbb{R}^d$. We assume we only have access to $\tilde{p}$, since the normalizing constant $Z$ may be intractable. For a given function $h : \mathbb{R}^d \to \mathbb{R}$, we wish to calculate the expectation under $p$, ie. $\mathbb{E}_p[h(X)]$.

In this work we make use of the **Stein Operator** (Gorham and Mackey, 2015), which for some function $g(x) : \mathbb{R}^d \to \mathbb{R}^d$ is defined as $\frac{1}{p} \nabla \cdot (p(x)g(x))$. This can be rewritten in the more useful form:

$$\mathcal{T}_p g(x) = g(x)^\top \nabla_x \log p(x) + \nabla \cdot g(x) \tag{1}$$

Notice that this depends only on the score function $\nabla_x \log p(x)$, which is independent of the normalizing constant $Z$. An important property of the Stein operator is that $\mathbb{E}_p[\mathcal{T}_p g(X)] = 0$, so long as $p(x)g(x) \to 0$ on the boundary $\partial V$, where $V$ is the support of $p(x)$ (This follows by applying the divergence theorem to evaluate the integral).

The Stein operator can be thought of as a device mapping a function $g(x)$ into one that has zero mean under $p$ (under mild conditions on $g$). There are several works which make use of the Stein operator and its properties (Gorham and Mackey, 2015; Ranganath et al., 2016; Grathwohl et al., 2020).

## 2.1. Control Variates

A control variate (CV) in our setting is a function $s_\theta : \mathbb{R}^d \to \mathbb{R}$, with parameters $\theta$, which satisfies the zero-mean property: $\mathbb{E}_p[s_\theta(x)] = 0$.

This allows us to construct a new estimator for the expectation as: $\mathbb{E}_p[h(X) + s_\theta(X)] = \mathbb{E}_p[h(X)]$. Since the CV has learnable parameters, they may be tuned to produce an estimator with smaller variance. In particular, the training objective for the CV can be formulated as:

$$\mathcal{L}_{\text{Var}} = \sum_i (h(x_i) + s_\theta(x_i) - \text{mean}(h(x_i) + s_\theta))^2 \tag{2}$$

The mean term is commonly replaced with a learnable constant $c$, so that the objective coincides with regressing the parametric function $-s_\theta(x) + c$ to the target $h(x)$ on samples $x_i \sim p$ (Oates et al., 2017; Wan et al., 2019; Sun et al., 2023). We refer to this as the "regression objective" for CV methods.

## 2.2. Zero-Variance CVs

Zero-Variance CVs use the earlier property of the Stein operator to construct flexible CVs (Assaraf and Caffarel, 1999). In particular, they define the CV as: $s_\theta(x) := -\mathcal{T}_p g_\theta(x)$. The benefit of this approach is that there are very few restrictions on $g$, and the corresponding space of CVs can be very flexible.

In particular, for a flexible parametric class $g$, they can minimize the variance to 0, which corresponds to the random variable $h(X) - \mathcal{T}_p g_\theta(X) = \mathbb{E}_p[h(X)]$, a constant. Rearranging this, it means that to obtain the optimal CV, we want it to solve the following differential equation for $g(x)$, known as *Stein's equation*, or sometimes, the *the fundamental equation* (Assaraf and Caffarel, 1999):

$$h(x) - \mathbb{E}_p[h(X)] = \mathcal{T}_p g(x) \tag{3}$$

Under mild conditions, this equation has a solution (though not necessarily unique) (Oates et al., 2017).

Existing works explore parameterizing $g$ in different ways, eg. with polynomials (Assaraf and Caffarel, 1999), in an RKHS (Oates et al., 2017), or with neural networks (Wan et al., 2019; Sun et al., 2023). These methods largely use the same variance minimization perspective (2) (in particular its regression variant). The key issue is that this objective requires samples from $p(x)$, which are often difficult to obtain. In theory the method should still be valid if the samples $x_i$ are not from the target $p$, since it is attempting to solve Eq. (3) using regression. However, the objective performs poorly in that case (this is also confirmed in our experimental section). This phenomenon can be better understood through the following thought experiment. If we fix $g_\theta$ and train the $c$ parameter to optimality on $x_i \sim r(x)$, it will converge to $c_0 = \mathbb{E}_r[h(X) - \mathcal{T}_p g_\theta(X)]$. If we then train with a flexible function $g$ with this obtained $c_0$, the function will attempt to converge to a solution such that $h(x) - \mathcal{T}_p g_\theta(x) = c_0, \forall x$. In other words, the regression objective often produces an estimate that depends on $r$, which is not desirable.

The neural network methods are especially sensitive to this and produce bad results when the number of samples from $p$ is small. Previous works attribute this to "overfitting"

(Wan et al., 2019; Sun et al., 2023). They propose remedies such as regularization terms (Wan et al., 2019) or meta-learning across multiple estimation tasks (Sun et al., 2023) to make better use of samples.

## 3. Method

Instead of taking the variance minimization perspective with the Stein equation (3), we can formulate other objectives to solve for $g$, not requiring samples from $p$. Once this is done, we can estimate $\mathbb{E}_p[h(X)] = h(x) - \mathcal{T}_p g(x)$ where the right hand side should be constant over $x$ at the solution. The key issue is that equation (3) already requires us to know $\mathbb{E}_p[h(X)]$, so we can't directly solve it.

There are two methods for proceeding, making use of the fact that the right-hand side and left-hand side differ only by a constant.

- We can take the gradient of both sides to obtain yet another differential equation:

$$\nabla_x h(x) = \nabla_x \mathcal{T}_p g(x) \tag{4}$$

Minimizing the norm of the gradient difference on a set of points (a **mesh**) $\mathcal{M}\{x_i\}$ yields the "**Grad Loss**" objective:

$$\mathcal{L}_{\text{grad}}(\theta) = \sum_{i=1}^n ||\nabla_x h(x_i) - \nabla_x \mathcal{T}_p g_\theta(x_i)||_2^2 \tag{5}$$

- We can alternatively take the difference of equation (3) on points $x$, $x'$, and rearrange to cancel out the constant:

$$h(x) - \mathcal{T}_p g(x) = h(x') - \mathcal{T}_p g(x') \tag{6}$$

If we consider minimizing this over a set of points $\mathcal{M}$, and a set of perturbed points $\bar{\mathcal{M}} = \{x_i + \epsilon_i\}$, we get the "**Diff Loss**" objective:

$$\mathcal{L}_{\text{diff}} = \sum_{i=1}^n ||\big(h(x_i + \epsilon_i) - \mathcal{T}_p g(x_i + \epsilon_i)\big) - \big(h(x_i) - \mathcal{T}_p g(x_i)\big)||^2 \tag{7}$$

If the perturbations are from a Gaussian $\epsilon_i \sim \mathcal{N}(0, \sigma^2)$, then (7) can be seen as a zero-order approximation for the gradients in (5) (Liu et al., 2020) (though it is a valid loss function for any perturbations).

The above methods technically introduce more solutions, for which $h(x) - \mathcal{T}_p g(x)$ differs by a constant. In other words $g$ may solve (4), but not (3). Such solutions would give us incorrect expectations. These extraneous solutions are eliminated by the boundary condition $p(x)g(x) = 0$ on $\partial V$. In other words, so long as $g$ is a solution to (4), and satisfies the boundary conditions, it will give us the correct expectations.

The "Grad Loss" objective (5) is expected to be better since the gradient naturally picks out the direction of maximal disagreement for the objective, while this needs to be randomly estimated in (7). However, it requires back-propagating through $\nabla \mathcal{T}_p g_\theta(x)$, a

$\mathbb{R}^d \to \mathbb{R}^d$ function. This will quickly become very expensive in higher dimensions, so (7) should be thought of a a cheaper alternative.

Our proposed method is essentially to use a neural network $g_\theta(x)$ to approximately solve the equation (4) or (6) everywhere in space. We can solve the associated losses on a mesh to achieve this. Notably, the mesh need **not** be sampled from the target $p$.

We refer to this method as ***Neural Stein Estimation*** (NSE). The overall NSE algorithm, for the loss (5), is presented in Figure 1.

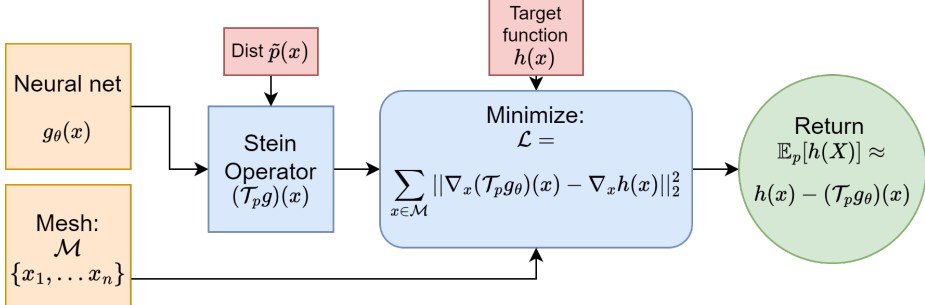

Figure 1: The Neural Stein Estimation (NSE) algorithm

This method yields a way to calculate expectations without sampling. In contrast to MCMC, it is more stable, and supplies a loss function with which we can monitor progress.

In contrast to control variate methods the $x_i$ doesn't need to be samples from $p$. This bypasses the issues with "overfitting" noted when using neural networks (Wan et al., 2019; Sun et al., 2023), since we are free to generate points in the mesh as we like.

The mesh still requires attention: namely it should capture "salient" features of the distribution $p$, such as modes, or boundaries of the support (if it's finite). Some examples are included in the appendix. In this sense, a mesh distribution $r(x)$ that places points on high-probability regions of $p$ is needed. This doesn't mean $r(x) = p(x)$; in particular, the mesh statistics (e.g., the mean and variance) need not be the same as those of $p$.

For the purpose of this work, we choose the solve (4) or with a neural network. We could have instead used a numerical method. The reason for preferring neural networks is that for future work, we'd like to *amortize* the procedure of learning $g$ across multiple different $h(x)$.

## 4. Experiments

We experimentally test the viability of the method (with loss (5) labelled "NSE (G)" and (7) labelled "NSE (D)") on a multivariate Gaussian distribution. We estimate the second moment (summed across dimensions) $\mathbb{E}[\sum_{i=1}^d X_i^2]$ for the multivariate Gaussian $\mathcal{N}(3, 5I_d)$. We compare to the CV baselines such as Control Functionals (CF) (Oates et al., 2017) and Neural Control Variates (NCV) (Wan et al., 2019), as well as to MCMC methods such as (unadjusted) Langevin Monte Carlo (LMC) (Rossky et al., 1978; Parisi, 1981), and Hamiltonian Monte Carlo (HMC) (Duane et al., 1987; Neal et al., 2011).

Figure 2 compares the moment estimates as dimension increases. The training mesh used is sampled uniformly from the hypercube $r(x) = \mathcal{U}[-10, 10]^d$, and a 4-layer neural

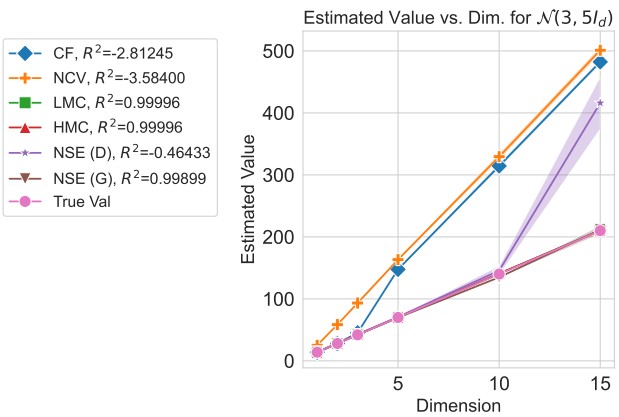

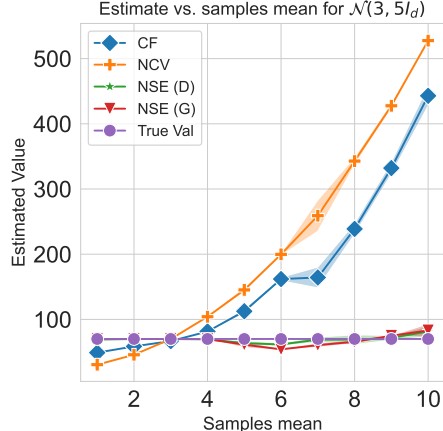

Figure 2: Estimation of $\mathbb{E}[\sum_i X_i^2]$ for different number of dimensions when $p(x) = \mathcal{N}(3, 5I_d)$.

Figure 3: Estimation of $\mathbb{E}[\sum_i X_i^2]$ for $p(x) = \mathcal{N}(3, 5I_5)$ for different values of the mesh mean.

network is used for NSE (and NCV). MCMC methods perform best overall, while among the methods which use the mesh, NSE with the gradient loss performs best, followed by NSE with the difference loss.

Figure 3 fixes the dimension to 5, and trains the CV methods and NSE on a mesh sampled from $\mathcal{N}(\mu, I)$, for varying $\mu$. In this case, we observe that the CV methods are less robust to changes in mesh compared to NSE. Note that these results for CV use the regression objective. Directly using the variance objective (2) produces different results (sometimes better and sometimes worse), which requires further investigation.

## 5. Conclusion

We presented *Neural Stein Estimation* (NSE), a method which approximately solves a differential equation to estimate the expectation of a given function $h(x)$ under a distribution $p(x)$. The method is stable, doesn't require sampling, and empirically demonstrates favourable properties compared to CV methods when they aren't trained on target samples. Some future directions to improve the work include:

- The method only estimates a single expectation, in contrast to MCMC. One possible route to address this is to amortize the procedure of learning $g$ across a set of expectations $\{h_i\}$. The ability to meta-learn control variates across different $h(x)$ (and $p(x)$) as in (Sun et al., 2023) suggests that this amortization may be practically achievable.

- The method currently works only for scalar-valued integrands, $h : \mathbb{R}^d \to \mathbb{R}$. There may be ways of efficiently adapting the method to vector valued integrands, which would allow the method to be used for gradient estimation tasks.

## Acknowledgements

This research was enabled in part by compute resources provided by Mila (`mila.quebec`), Calcul Québec (`calculquebec.ca`) and the Digital Research Alliance of Canada (`alliancecan.ca`). The authors are also grateful for the funding from CIFAR and Samsung.

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

## Appendix A. The Choice of Mesh

We start with some experiments clarifying the role of the mesh, and to better understand when our method fails. Specifically, we evaluated the method on one-dimensional distributions, specifically the Mixture of Gaussians ($\sim 0.5\mathcal{N}(-10, 3^2) + 0.5\mathcal{N}(10, 3^2)$), estimating the mean $h(x) = x$, and the Exponential distribution ($\sim Exp(1)$), estimating the second moment $h(x) = x^2$. The mesh is a grid of equally spaced points over some interval.

Figure 4 contains two examples demonstrating the impacts of both inadequate and adequate mesh selections. For the MoG distribution, the top left figure shows the result of using an inadequate mesh, yielding an estimated expectation $E[X] \approx 10$. In contrast, the bottom left figure, with an adequate mesh, accurately estimates $E[X] \approx 0$. The reason for the failure is that the mesh covers only 1 mode of the distribution $p$. The differential equation is thus solved assuming $p \approx \mathcal{N}(10, 3^2)$, producing an estimate of its mean, since this is the only information the loss function observes.

Similarly, for the Exponential distribution, the top right figure with an inadequate mesh results in $E[X^2] \approx 11$, while the bottom right figure, using an appropriate mesh, correctly estimates $E[X^2] \approx 2$. Notice that for this problem, one of the boundaries is $x = 0$. Here the issue is that the loss function doesn't observe the differential equation on an interval starting from the boundary. For meshes which do, the neural network learns a smooth solution respecting the boundary condition, producing accurate expectations. Otherwise the neural network can learn a different solution which violates the boundary conditions. Note that this occurs even if we fix the architecture of the net to enforce the boundary

condition (for instance here, outputting $g(x) = x\phi(x)$, with neural net $\phi$). The reason is that the neural net is very flexible and can attain different solutions on different intervals, depending on where it sees data.

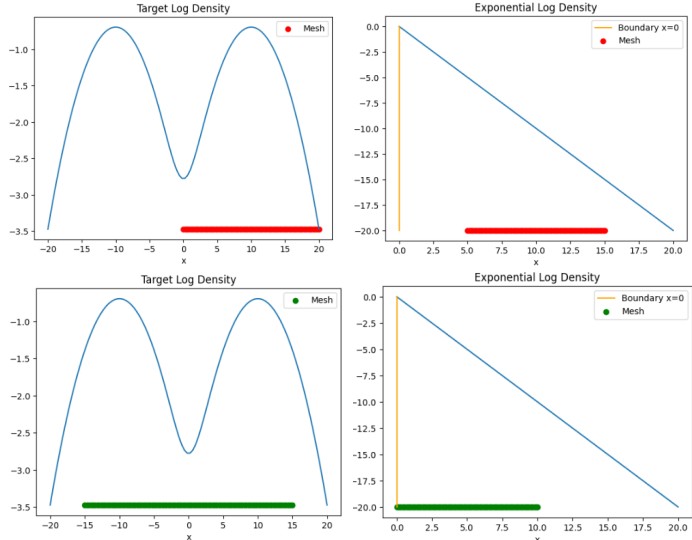

Figure 4: Impact of mesh selection on 1D MoG and Exponential distributions. Top Left: MoG with inadequate mesh: $E[X] \approx 10$. Bottom Left: MoG with adequate mesh: $E[X] \approx 0$. Top Right: Exponential with inadequate mesh: $E[X^2] \approx 11$. Bottom Right: Exponential with adequate mesh: $E[X^2] \approx 2$.

From these examples we can observe that:

- The mesh should cover representative regions of the distribution $p$ (for instance, multiple modes). In this sense, the challenges from multimodality persist with NSE, but in a different way from MCMC.

- (For non-asymptotic boundaries), the mesh should cover the boundary $\partial V$ of the support of $p$.

