# OpenReview forum: "Estimating Expectations without Sampling: Neural Stein Estimation"
_approximateinference.org/AABI/2024/Symposium — AABI 2024_

### Official Review · Reviewer_adFS · 2024-04-15
**Review of paper #25: Interesting variation on a well studied subject**

**Rating:** 7
**Confidence:** 4

**Review:**

The paper considers estimating an expectation of a function h(x), where x is governed by a distribution p(x).  The paper is clearly written and puts the work in the context of sampling methods.  A neural Stein estimation (NSE) method is developed and tested for a normal distribution with growing dimension.  The paper takes the neural approach as a potential first step to obtaining more general solutions for the expectation estimator, so in this sense the work opens a potential avenue for advances harnessing learning.

A strong motivation for the method is computational complexity savings, although the paper doesn't make comparisons.  The NSE requires a mesh sampler, rather than say a random MCMC sampler; both are required to sufficiently explore the distribution p(x) and the tradeoffs aren't clear from a computational perspective.  The appendix makes it clear that the choice of mesh is crucial for interesting distributions (multi-modal), and so there appears to be a kind of explore-exploit tradeoff needed here or good prior knowledge.  So, in practice, it doesn't seem quite clear how the complexity is to be judged.

This basic assumptions for this class of methods are somewhat conflicting, because it is assumed that the score function is available.

The method of scoring as a means of local optimization is classic, and relates to the CRB which is a local bound.  In the NSE method it isn't quite clear to the reviewer how this relates to local vs global optimality in the estimation of the expectation.

The example is interesting, expanding to d = 20 dimensional problem, but only for a Gaussian case.   Of course, what constitutes a 'high' dimension is very relative, so perhaps this is high from a Bayesian methodology approach, yet not so high for learning-based methods in general.

---

### Official Review · Reviewer_71sP · 2024-04-24
**Interesting work, and clear and smooth expression**

**Rating:** 7
**Confidence:** 4

**Review:**

This paper uses the Stein equation, NN and control variate to estimate the expectation without sampling (with mesh). It is an interesting work with clear and smooth expression.

Several suggestions for modification:
1. Can the experiment include more setup details(NN config)? Other metrics, such as time of computing consumption, may be included;
2. Is the mesh like a prior? Although the appendix gives some explanation, it still needs more attention maybe.

The results of containing NN are better, are they related to NN's generalization? Maybe you also can consider the NN with uncertain qualifications (more robust).

Hope to see the open source of your project.

---

### Official Review · Reviewer_2s1L · 2024-04-24
**more question than paper**

**Rating:** 5
**Confidence:** 3

**Review:**

I have an impression, that paper somewhat re-introduce ideas from, for example, https://arxiv.org/abs/1803.10161.
However, instead of using local MCMC movements the idea is to use grid. The problem with the grid which I see, is exactly for MCMC approaches exists at the first place. With dimensions bigger that 5 and density not unimodal the size of uniform grid required to evenly cover the space will be exponentially big. So, if we start to think on how to put the grid more dense in places where density is more "typical" we will invent some type of MCMC proposal.

This doesn't mean that regular grids can not be used. However, underlying mechanism is not clear. One idea which is typical for community is tensoring the grid with Tensor Train, not sure if this applicable directly here, but authors can try.

---

### Official Review · Reviewer_2gq7 · 2024-04-24
**Neural Stein estimation estimates expectations without sampling (but works best with sampling).**

**Rating:** 7
**Confidence:** 2

**Review:**

This submission considers the problem of calculating expected values for a function $h$ of a random variable with density $p$. The method builds on the control variate approach combined with Stein's operator to allow for a flexible class of control variate distributions $g$. Previous approaches to this problem have depended on generating samples from $p$, which may be computationally expensive. This can be sidestepped by either taking the gradient of both sides of Stein's equation, yielding another differential equation (Grad Loss), or the solution can be approximated by minimizing the objective over the difference between a set of points and their perturbation (Diff Loss). The approach then consists of using a neural network to approximately solve the resulting (Grad/Diff Loss) equations everywhere in space via approximation on an (in principle) arbitrary grid.

This method is demonstrated on a toy example of estimating the second moment for a multivariate Gaussian as the dimension ranges from 1 to 20. The example also illustrates results for standard Monte Carlo estimation along with control functionals (CF) and neural control variates (NCV). For this simple example, of the methods not based on sampling from $p$ the Grad Loss approach provides the best approximation, followed by Grad Diff.

While the proposed Grad Diff/Loss approaches appear favorable in the toy example, as the authors acknowledge to capture "salient" features of the distribution $p$ a mesh $x_i \sim p$ would be ideal; however, that would be equivalent to sampling from $p$. It is intriguing the proposed methods seem to perform well on the toy problem, but it is impossible to say whether this is a general feature of the approach or simply reflects the problem at hand. That said, the method considers a general problem and discussions with other researchers as a poster at AABI may lead to additional progress.

Overall the submission is clearly written, though figure 2 would benefit from a larger pointsize overall. There also appears to be a typo in the title to figure 2, which I believe should be "N(1, 5²I)".

---

### Official Review · Reviewer_1xwU · 2024-04-25
**Theory is fine, but the method is unconvincing. However, perhaps the workshop can aid in developing the work further.**

**Rating:** 6
**Confidence:** 5

**Review:**

This paper is about using Stein's identity to estimate expectations with respect to probability distributions. They design two objective functions to train a neural net that solves some surrogate/proxy for Stein's identity. Thereafter, they use the neural net to estimate the expectation of interest. While the theory behind it is interesting and sound, I am not convinced that the approach the authors take the compute the expectations is the best use of the theory. From a computational standpoint, the method is quite expensive relative to the task it solves. In theory, one must solve this differential equation each time one wishes to obtain a new expectation.

The authors claim the method is more robust ; but what if the mesh does not properly capture the target distribution? Other Stein's identity-based sampling schemes suffer similar issues as MCMC in situations with multiple modes. That is, if the initial condition is not well-placed, certain modes may be near impossible to discover. I would imagine similar issues may appear here as well? Is there something inherently better about working with expectations directly? As is, I don't see an advantage of this approach over something like SVGD or some neural version of it, especially since once one has samples, one can estimate expectations for a large class of functions.

I found the discussion on control variates somewhat unnecessary, especially since you could start from stein's identity in equation (3) and say we wish to solve this equation for g, where g is parametrized with a neural net. The discussion about control variates is relevant if one does Monte Carlo afterwards, but since the goal of the paper is to obtain zero-variance control variates and avoid sampling altogether, so the discussion seems unnecessary.

However, I do recommend the paper for participation in the workshop as it may help further develop the work in more fruitful directions.

---

### Official Review · Reviewer_523P · 2024-04-27
**Paper Review**

**Rating:** 6
**Confidence:** 3

**Review:**

# Overview
In this work, the authors introduce Neural Stein Estimation, an approach for approximating integrals without resorting to drawing samples from the target distribution.
They propose using a neural network as a control variate term and consider two loss functions for optimising it, resulting in estimates of the integral in question without requiring access to samples from the base distribution.
They test the required approach on a simple toy problem and compare this to some existing baselines.

# Quality and clarity
In my view, the quality and clarity of the paper could be improved.
For example, after reading the paper, it is not fully clear to me whether the proposed approach yields unbiased estimates of the integral in question.
I also thought that the method was not explored in sufficient depth, neither theoretically nor experimentally (see comments on experiments and guarantees below).
Overall however, the motivation of the paper was clear.


# Originality and significance
The proposed method appears original, and different to existing methods in the literature, at least in my view.
While the results in the paper are somewhat limited, the toy experiment considered by the authors shows some promise, and the method could prove significant if further developed.


# Comments and suggestions for improvement

__Comparison to Stein Variational Gradient Descent:__
The proposed method seems somewhat similar to Stein Variational Gradient Descent (SVGD).
There are clear differences in the sense that, for example, SVGD approximates a distribution rather than an integral.
In this sense, SVGD could be used to tackle the same problems as the proposed method (by first estimating the distribution and using the estimate to approximate the integral).
It would be interesting and insightful to compare against SVGD, which, similarly to the proposed method, does not require samples from the base distribution, and, unlike the proposed method, does not require the choice of a mesh (though a kernel function still needs to be chosen).

Qiang Liu, Diling Wang, Stein Variational Gradient Descent: A General Purpose Bayesian Inference Algorithm, 2016.

__Comparisons to MCMC and / or timings:__
The authors motivate their proposed algorithm by pointing out that MCMC methods can be expensive to run and biased.
However, state-of-the-art MCMC methods, like Hamiltonian Monte Carlo (HMC) and its variants, can be very effective in practice.
I think that a comparison between the runtimes and biases of such MCMC methods and the proposed method (even a rudimentary comparison) would be useful.

__Picking an appropriate mesh:__
It seems that picking an appropriate mesh is important for the proposed method to work well.
Since the optimal mesh will in general depend on the problem at hand, one can imagine that picking a good mesh may be difficult to achieve manually.
Have the authors considered any approaches that would enable the mesh to be automatically tuned to the problem at hand?

__Discussion on guarantees:__
Currently, there is little discussion on the paper regarding the guarantees of the proposed method.
For example, does the proposed method yield unbiased estimates or not (I presume it does not, but I might be wrong about this)?

__Experiments are somewhat thin:__
In my view the experimental section in the paper is very thin.
Given that the method is not explored in significant depth from a theoretical perspective, a more thorough set of experiments, e.g. on different toy distributions and integrands would be beneficial.

__Including pseudocode:__
In my view, including some pseudocode that lists the computations carried out by the algorithm concretely would help the clarity of the paper.
Figure 1 is useful, but a slightly more formal description would be beneficial also.

---

### Meta-Review · Area_Chair_rgA6 · 2024-05-24

**Recommendation:** Accept (Poster)
**Confidence:** 3

**Metareview:**

This paper proposes a method for estimating expectations without drawing samples and instead relying on Stein's identity and neural network-based control variates. Most of the reviews recommended marginal acceptance, with one suggesting marginal rejection based on a lack of originality/missing related work discussion. The "acceptance" reviews themselves were mixed, with some disagreement about the paper's strengths and weaknesses. Nonetheless, the paper does seem to be clearly written and of interest to the community; thus, I recommend acceptance. I would encourage the authors to expand the related work discussion and experimental evaluation for the camera ready.

---

### Decision · Program_Chairs · 2024-05-27

Accept